# Individual- and group-level network-building interventions to address social isolation and loneliness: A scoping review with implications for COVID19

Reza Yousefi Nooraie[1]☉*, Keith Warren[2]☉, Lisa A. Juckett[3]☉, Qiuchang A. Cao[2]☉, Alicia C. Bunger[2]☉, Michele A. Patak-Pietrafesa[2]☉

1 Department of Public Health Sciences, University of Rochester, Rochester, New York, United States of America, 2 College of Social Work, The Ohio State University, Columbus, Ohio, United States of America, 3 School of Health and Rehabilitation Sciences, The Ohio State University, Columbus, Ohio, United States of America

☉ These authors contributed equally to this work.
* Reza_Yousefi-nooraie@URMC.Rochester.edu

**Data Availability Statement:** All relevant data are within the paper and its Supporting information files.

## Abstract

### Purpose

Worldwide mandates for social distancing and home-quarantine have contributed to loneliness and social isolation. We conducted a systematic scoping review to identify network-building interventions that address loneliness and isolation, describe their components and impact on network structure, and consider their application in the wake of COVID19.

### Methods

We performed forward and backward citation tracking of three seminal publications on network interventions and Bibliographic search of Web of Science and SCOPUS. We developed data charting tables and extracted and synthesized the characteristics of included studies, using an iteratively updating form.

### Findings

From 3390 retrieved titles and abstracts, we included 8 studies. These interventions focused on building networks at either individual- or group-levels. Key elements that were incorporated in the interventions at varying degrees included (a) creating opportunities to build networks; (b) improving social skills; (c) assessing network diagnostics (i.e. using network data or information to inform network strategies); (d) promoting engagement with influential actors; and (e) a process for goal-setting and feedback. The effect of interventions on network structures, or the moderating effect of structure on the intervention effectiveness was rarely assessed.

**Funding:** The author(s) received no specific funding for this work.

**Competing interests:** The authors have declared that no competing interests exist.

## Conclusions

As many natural face-to-face opportunities for social connection are limited due to COVID19, groups already at risk for social isolation and loneliness are disproportionately impacted. Network-building interventions include multiple components that address both the structure of individuals' networks, and their skills and motivation for activating them. These intervention elements could be adapted for delivery via online platforms, and implemented by trained facilitators or novice volunteers, although more rigorous testing is needed.

## Introduction

The new norm of social/physical distancing and home quarantine after the COVID19 pandemic is contributing to the increase of social isolation and loneliness [1], which may have a significant impact on the physical and mental health of vulnerable populations [2]. Recent lockdowns due to COVID19 have brought the problems of social isolation and loneliness to the forefront of public attention [3], but these problems are not new, as loneliness is as old as human history. There is evidence that both social isolation and loneliness have increased in the United States in recent years [4–7].

Loneliness and social isolation are complex, multilevel phenomena. Social isolation refers to the lack of social contacts and engagement [8], whereas loneliness reflects subjective dissatisfaction with the quality or quantity of social contacts [9]. They are independently associated with physical illness, mental illness, and mortality [10]. Everyone can experience varying intensity and duration of loneliness and/or isolation at a certain point in life [11]. However, some populations such as those who are older, people who are LGBT or who have cognitive disabilities, are disproportionately affected by social isolation and loneliness, while gender, health and income also play a role [12–14].

### Loneliness/Isolation intervention as network-building interventions

Interventions to address isolation/loneliness generally aim to improve the quantity and quality of social relations with existing or new support individuals and groups. Network-building interventions are deliberate efforts to change social networks of individuals [15]. Social network analysis (SNA) is an important approach to assess how interventions addressing loneliness/isolation change social networks. SNA is a well-established approach that focuses on the relational patterns between network actors [16, 17] rather than considering them as separate units. In other words, SNA captures the interdependencies among network actors, whereas conventional research methodologies assume independence among network participants [18].

A popular approach to study the network outcomes of loneliness/isolation interventions is assessing the change in the size of personal networks [19], or individuals' evaluation of support networks [20, 21]. Many existing social network tools, including Berkman Syme Social Network Index [22] and Lubben Social Network Scale [23], ask about the number of individuals in each important social role (family, friends, colleagues, etc.), and usually provide a single score. Some researchers ask about the number of important individuals in respondents' social networks [24], create numerical scores based on a mix of the number and quality of social relations [25], or ask respondent's evaluation of the number of people who can help each other in the community [26].

In the following section, we argue that the number of ties does not fully capture the relational complexity of social networks in which one is embedded.

## Network structure and well-being

Several aspects of network structure influence individuals' perception of loneliness and support. Here we provided a few examples:

**Centrality** represents the prominence of an actor in the network. The simplest measure of centrality, degree centrality, is the number of network connections an individual has (e.g. the number of friends, or family members) [27]. Degree centrality is one of the most commonly measured indicators of network structure, although authors do not necessarily refer to a count of connections as a measure of centrality [19]. In the case of social support, degree centrality could either be indegree (the number of people who offer you social support or who name you as a friend) or outdegree (the number of people to whom you offer social support or whom you name as a friend). Indegree centrality is positively correlated with emotional support [28]. However, a variety of other measures of centrality are associated with well-being. For example, betweenness centrality, the extent to which an individual is on paths between other members of the network, and closeness centrality, the extent to which short paths connect the individual and other members of the network, are positively correlated with measures of wellness [29]. Indirect connections also matter. Being connected to more people indirectly (through others) decreases the likelihood of depression [30].

The **density** (connectedness) of networks affects well-being. Compared to degree centrality which is about the number of relations from or towards ego (respondent), density is about the overall connectedness of personal network, which should include relationships among network members not including ego. Density of personal networks was shown to predict loneliness in college students, particularly in men [31, 32]. However, some studies failed to show the association between density and subjective loneliness [33, 34], or proposed that the effect of density of personal networks on life satisfaction is moderated by an individual's personality (as individuals may differ in terms of satisfaction by embedding in denser networks) [35].

**Reciprocity** (bi-directionality) of social relations matters for several reasons. Qualitative studies have found that older individuals have an easier time accepting social support when it is offered as part of a reciprocal relationship, that is, when they feel that they have given (in the case of their children) or are giving help to the person who is helping them [36, 37]. Individuals who perceive reciprocity in their relationship with their best friends feel less lonely [38]. There is experimental evidence that reciprocity is a key factor in building trust in networks [39]. Reciprocal imbalance in relationships (over-benefiting and under-benefiting) may lead to mental distress and less satisfaction with relations [40]. All of these findings suggest that reciprocity may be one key to maintaining relationships once they are established.

**Network clustering**, or the degree to which groups of three individuals connect completely with each other, is another important feature. In a longitudinal analysis of a large population-based study, Cacioppo and colleagues (2009) found that loneliness occurs in clusters, particularly clusters of individuals that are peripheral to the social network [41]. People in more clustered networks tend to be healthier [29]. Experiments find that the search for cooperative partners produces clustering, suggesting that clustered networks play a role in maintaining cooperation in groups [42, 43]. Of course, the influence that clustered networks have on their members has a downside. DiFonzo et al (2014) find that clustering increases the social influence of the group but also increases the tendency of groups to strongly differentiate themselves from others [44].

Based on the abovementioned evidence, we argue that interventions addressing loneliness/ social isolation may affect the structure of social networks beyond merely increasing the number and frequency of social relations to ego. They may change the reciprocity of relations, may affect formation of denser clusters, or may even selectively affect some regions of one's social network (e.g. only improving the quality of intimate relations, or leading to bridging ties to new clusters). Given the need for more in depth analysis of the structural targets of interventions for loneliness/isolation, we conducted a systematic scoping review of studies assessing the effect of interventions to address loneliness and social isolation on the structure of social networks. We aimed to map the types and components of these interventions and methods/ metrics of assessing structural changes in social networks. We evaluated how these interventions can be adapted to promote connectedness in the context of societies after COVID19.

## Methods

We followed PRISMA-ScR guidelines for scoping reviews in the conduct of the literature review, data extraction/charting, and synthesis [45].

### Literature search

Our search strategy involved forward/backward citation tracking of three seminal network intervention publications, Valente (2012) [15], Valente et al. (2015) [46], and Latkin & Knowlton (2015) [47], complemented by a bibliographic search in Web of Science and SCOPUS, performed in October 2019 (search strategy in S1 Appendix). To capture articles about relevant interventions for social isolation and loneliness that might not have been described using network intervention keywords, we also conducted a hand search of seminal reviews on the topic [19–21, 48]. Articles identified through this search strategy were imported into the web-based review program, Covidence [49], before undergoing title/abstract review.

### Study selection

To establish consistency in the study selection and data charting process, all authors completed a trial screening process in pairs (three sets of pairs total). Authors met biweekly from March 2020 –July 2020 to discuss points of disagreement and achieve consensus pertaining to title/ abstract and full-text study inclusion. We included studies that intentionally aimed to change aspects of social networks to address isolation/loneliness, measured the network structure as study outcomes and/or used them to inform interventional strategies, and were available in English. We excluded studies that only reported the number of individuals one is connected to (network size) or only provided aggregate measures of quality and quantity of relations (such as Berkman Syme Social Network Index and Lubben Social Network Scale).

### Data extraction and synthesis

A data-charting form was jointly developed by the reviewers to determine variables to extract. Pairs of reviewers independently extracted a calibration set of studies. Once the team demonstrated consensus and consistency in data charting on the calibration set, the information in each included study was extracted by one author. The group regularly met to discuss and resolve uncertainties and potential disagreements, and continuously updated the data-charting form. Categories included on the data charting form were as follows: author/year, sample, study setting, design, characteristics of network actors, the nature of network relations, characteristics of the intervention(s), theoretical underpinnings of the intervention(s) according to the authors, measures of network change and other outcomes, effectiveness of the intervention

according to the authors, and implications. Result syntheses consisted of three steps: 1) descriptive summary describing details about study design, 2) thematic analysis for the categorization of network interventions, and 3) consideration for how these interventions could be adapted to contexts with limited physical and in-person contact opportunities, as manifested by COVID19 restrictions. We did not conduct quality appraisals of included studies, which is consistent with PRISMA-ScR guidelines given the exploratory nature of scoping review methodology and the heterogeneity of studies included in scoping reviews.

## Results

### Literature search and selection process

Of our initial 3390 references, we assessed the full texts of 233 articles, of which 17 studies were about the effect of interventions on social networks to address loneliness and isolation, of which 9 studies were excluded, as they either only reported the number of individuals one is connected to or developed single aggregate measures for quality and quantity of relations without providing information about the composition of social network and/or detailed information about relations to particular social groups/roles. We included eight studies that measured network structure before and after the intervention and/or incorporated network analysis as a component of the intervention (Fig 1). Three studies assessed interventions focusing on *individual-level network building*, and four studies assessed *group-level network building interventions*. Kasari et al. (2016) compared an individual-level with a group-level intervention, and consequently was included in both sections [50].

### Individual-level network interventions

The interventions in this category generally aimed to help the individuals with limited social connections (e.g. older adults and children and adults with intellectual, mental, and physical disabilities) strengthen/activate existing social relations or engage in activities to facilitate new relations or strengthen current ties (Table 1).

**Study designs.**  Two studies were small case reports, small uncontrolled studies, or qualitative studies. Kasari et al. (2016) conducted an RCT comparing the individual-level intervention (matching subjects with typically developing peers in social activities and games) and a group-based social skills training (explained in the next section) [50]. Band et al. (2019) published the protocol for an upcoming pragmatic RCT of an online platform to develop personal maps, and connect adults to local and online activities and resources, compared to a wait-list control [51].

**Intervention components.**  Interventions usually included elements of motivational training, reflection, and goal setting.

The interventions in two studies included assisting the participants to develop personal network maps, and reflect on their structure. In Band et al. (2019) [51] and Osilla et al. (2016) [52], facilitators would help participants develop their personal maps, through an online platform in Band et al. (2019) [51], and a computer-assisted face-to-face discussion in Osilla et al. (2016) [52]. The online interface in Band et al. (2019) [51] provided concentric circles representing tie strength, and various social roles including individuals, groups, and pets. In Osilla et al. (2016) [52], the process included a structured network interview, followed by discussions on visualizations. In Osilla et al. (2016) [52], the participants also identified relationships between network members, also known as "alters" (whether alter 1 knows alter 2), but in Band et al. (2019) [51], the questions were all about the relationships with ego (the focal actor), with no mention of alter-to-alter relations.

In three studies, the intervention included facilitation of social contact and exposure, either through connection to local activities as a general opportunity for network building [50, 51,

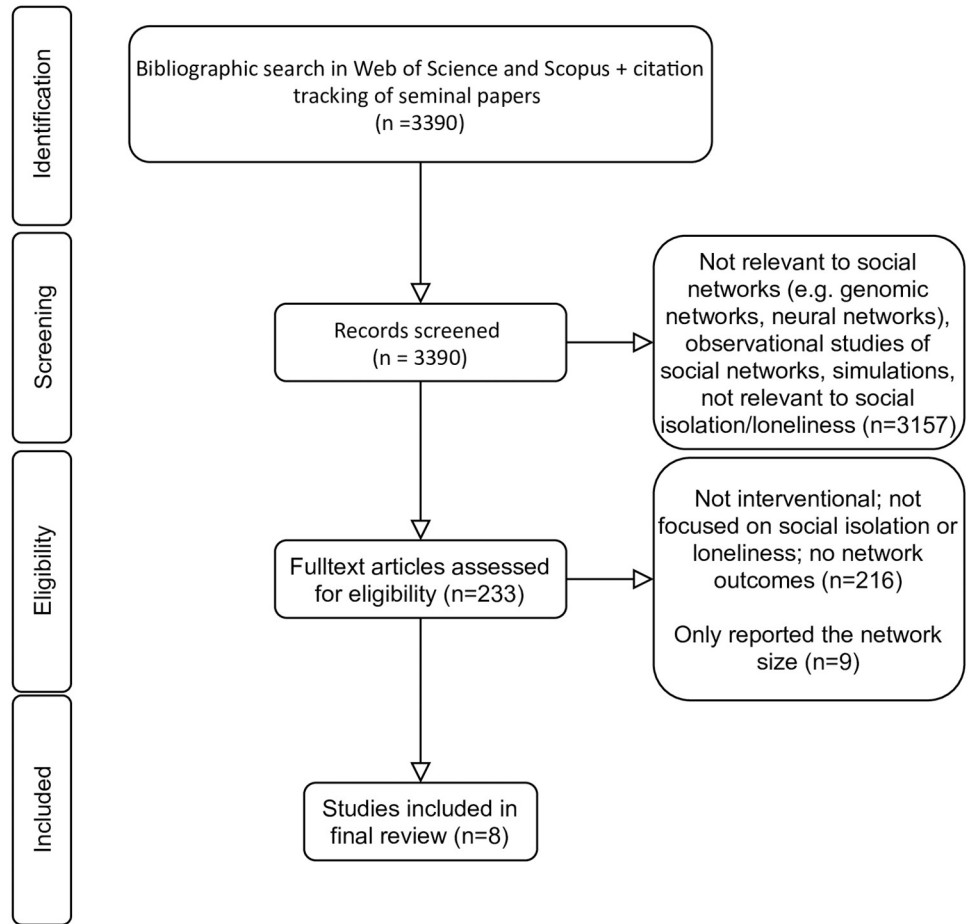

**Fig 1. PRISMA flow diagram of the study selection.**

53], or engaging in social activities with natural peers (or friends with no disabilities) [50]. In one study the intervention also included social skills training [53]. Two studies also included patients' family or support staff in the intervention [52, 53]. All studies described interventions consisting of several sessions lasting multiple weeks, to provide enough time for participants to reflect on their networks and make changes over time.

Two studies examined social network interventions that were administered in-person. In Band et al. (2019) [51] the intervention is an online platform to develop personal maps, and connect adults to local and online activities and resources. Similarly, Osilla et al. (2016) [52] assessed the effect of a computer-assisted motivational network intervention, in which the process of data collection and presentation of network maps was done on a tablet.

**Outcomes.** Two studies used sociometric network surveys to develop network structures and how they changed over time [50, 53]. Both studies calculated network salience/inclusion scores at each time point, using a method developed by Cairns & Cairns (1994) [54]. The score for each individual is the ratio of the number of times they were identified by others belonging to a group to the largest baseline score in class at baseline. Both studies showed that the intervention resulted in increased socialization of the participants (measured by the time spent with others). However, the change in network inclusion scores were modest and transient.

**Table 1. Study characteristics, elements of network-building interventions, and main outcome measures.**

| Study | Study design & Sample Characteristics | Setting | Network Ties | Network-building intervention | Network Analysis procedures & Measures | Intervention Effectiveness |
|---|---|---|---|---|---|---|
| | | | | **Individual-level interventions** | | |
| Band et al. (2019) [51] | **Design:** Pragmatic RCT with a wait-list control **Sample:** Adults (18+) who are isolated or at risk of loneliness; n = 394 | **Study:** Community organizations in two areas in the UK. | Ties with family members, friends, acquaintances, healthcare professionals, local groups and pets | **Intervention level:** Mixed: Network assessment was conducted online, in-person, or via phone. Individuals were connected to online resources. **Intervention summary:** Generating Engagement in Network Involvement (GENIE): Online social networking tool to create opportunities for social involvement through social network mapping, tailoring of preferences and linking users to valued resources and activities, delivered by trained facilitators. Facilitators helped participants develop personal support maps using concentric circles representing tie strength, and various social roles including individuals, groups, and pets [no mention of alter-to-alter relations] | **Measures:** SF-12 Mental Health and Physical Health Composite scores Loneliness, social isolation, well-being, Quality-adjusted life years, cost, Capability well-being, Collective efficacy in Network scale, engagement in new activities **Analysis:** Social network composition change (only in intervention arm) [since no alter-to-alter relations are collected, the composition will likely only refer to ego's relations] | Study in-progress (study protocol published) |
| Locke, et al. (2019) [53] | **Design:** Case report, feasibility pilot **Sample:** Elementary school students (grade 1–5) with Autism Spectrum Disorders; n = 4 | 2 public elementary schools in an urban district (85% of students are racial minorities & 70% of students receive free/reduced price lunch) | Observed playground social interactions & in-class friendship ties | **Intervention level:** In-person **Intervention summary:** Remaking Recess is an adult-facilitated intervention to support social engagement of children with ASD when at recess. The intervention includes observing if children need additional support to engage with their peers, follow children's lead, strengths and interests, providing direct instruction on social skills and games and activities to scaffold engagement, creating opportunities for reciprocal interactions, sustain engagement within an activity, coaching through difficult peer situations, work with typically developing peers to engage children with ASD, and eventually fading out supports to promote independence. | **Measures:** Sociometric survey of friendship networks within each class about students' social network and their perception of "who hang out together". **Analysis:** Centrality of students (number of times the student identified by others as belonging to a group) was used to calculate social network inclusion score (individual's centrality at each time divided by highest baseline centrality in class). | Decrease in solitary playground time; Increase in joint engagement with peers. Social network inclusion scores initially increased but declined at 6 week follow-up. |
| Osilla et al. (2016) [52] | **Design:** Qualitative feasibility and acceptability assessment **Sample:** Formerly homeless adults people with histories of AOD and HIV risk behaviors; n = 11 (8 men: 7 African American, 1 Latino/ Hispanic; 3 Women: 2 African American, 1 Latina/Hispanic) | Organization that provides permanent supportive housing in Los Angeles County | Social interactions | **Intervention level:** In-person (computer assisted) **Intervention summary:** Computer-assisted Motivational Network Intervention: Across 4 sessions (2-week intervals), participants complete a survey on drinking and unprotected sex and a name generator network survey followed by motivational discussions on the network map, the pros and cons of the patterns in the map, opportunities to discuss positive behavior change and goal setting for the following week. | **Measures:** No structural measures of personal networks reported **Analysis:** Qualitative study | Residents reported that the intervention was helpful in discussing their social network, that seeing the network maps was more impactful than just talking about their network, and that the intervention prompted thoughts about changing their AOD use and HIV risk networks. |

*(Continued)*

**Table 1.** (Continued)

| Study | Study design & Sample Characteristics | Setting | Network Ties | Network-building intervention | Network Analysis procedures & Measures | Intervention Effectiveness |
|---|---|---|---|---|---|---|
| Kasari et al. (2016) [50] | **Design:** RCT, comparing SKILLS (group social skill training) and ENGAGE (peer-matched social activities) **Sample:** Children (ages 6–11 in grades 1–5) with Autism Spectrum Disorder (IQ >65); n = 148; More than 50% were racial/ethnic minorities | Elementary schools | Friendship ties | **Intervention level**: In-person groups in schools **Intervention summary**: SKILLS: 16 sessions of group participation in interactive classes on social skills training: Being a Social Detective; Greetings and Goodbyes; Body Talk (Nonverbals); Humor; Conversation; Dealing with Teasing; Perspective Taking; Dealing with Emotions; and Friendship Tips. ENGAGE: Children with ASD and their typically developing peers (nominated through friendship survey or teacher nomination) engage in 16 sessions of group play activities, collectively establishing a daily schedule in order to encourage cohesiveness, conversational exercises, structured games, free play, improvised storytelling, and music. Peers were encouraged to take leadership of their own groups with supervision from adults as needed. | **Measures**: Sociometric survey of friendship networks within each class about students' social network and their perception of "who hang out together". **Analysis**: Centrality of students at class (number of times the student identified by others as belonging to a group) was used to calculate social network Salience score (individual's centrality at each time divided by highest baseline centrality in class). | All children significantly increased percent of time spent engaged with peers from pre- to post-treatment. Those in the SKILLS group improved significantly more than those in the ENGAGE group (significant treatment effect). Effect of treatment did not vary by site. No significant change in network salience scores over time however, there were modest overall improvements in both groups. |
| | | | | **Group-level interventions** | | |
| van Asselt-Goverts, et al. (2018) [55] | **Design:** Case report, qualitative assessment **Sample:** Adults with mild to borderline intellectual disabilities and their support workers; n = 5 | Organization providing support for people with intellectual disabilities in Netherlands | Ties to family, friends, neighbors, colleagues, acquaintances & professionals | **Intervention level**: In-person **Intervention summary:** A 7-session semi-structured group training to strengthen or expand the networks of participants. Sessions were facilitated by experienced trainers and focused on talents and interests, network, neighborhood, wishes and dreams, plans for a supporters meeting, and evaluation. Exercises included role playing and making a personal map. Facilitators helped participants develop personal support maps using concentric circles representing tie strength, and various social roles including family, friends, neighbors, colleagues, other acquaintances, and professionals [no mention of alter-to-alter relations]. For each network member, the participants scored the frequency of contact, affection, types of support, and preferences. Participants re-evaluated their maps after the intervention. | **Measures**: No structural measures of networks were reported. **Analysis**: Info gathered about each individual's network size and the frequency and quality of relationships. Data were analyzed to look for increases in size, frequency, and quality of relationships. | The network results were presented for individual participants and varied by individuals. In qualitative analysis, participants reported decreased loneliness and increased awareness, competence, autonomy and participation. |

*(Continued)*

**Table 1.** (Continued)

| Study | Study design & Sample Characteristics | Setting | Network Ties | Network-building intervention | Network Analysis procedures & Measures | Intervention Effectiveness |
|---|---|---|---|---|---|---|
| Gesell et al. (2013) [58] | **Design:** Uncontrolled pre-post test **Sample:** Parents of children at risk for obesity; n = 11 | Community recreation center in Nashville, TN | Advice and discussion ties with group members outside of training sessions | **Intervention level:** In-person group meetings **Intervention summary:** GROW intervention: The network-building intervention involved 12-week group skill building sessions in which social network diagnostics were used to create an action plan and recommendations for each group and its leader and a menu of action steps. Group sessions functioned to establish a strong group identity through developing and working toward a shared common goal and group identity. Group social networks were restructured through strategic pairing of isolates with highly connected group members, calling isolates in groups and promoting their participation, pairing non-reciprocated links, bringing triads together, and pairing members from different subgroups in small group activities; re-assigning members to prevent formation of silos. | **Measures:** Network diagnostic tool involved identification of isolates and components/ subgroups, and calculation of degree, density, reciprocity, transitivity, centralization, and average of inverse distance. **Analysis:** descriptive presentation of network measures at week 4 and 12. Density was compared using boot-strapped t-test. | Perceived cohesion increased (non-significant); Number of advice and discussion partners increased over time (non-significant); Advice network density increased significantly; Number of isolates remained stable; Subgroups decreased (integrated into network); Centralization increased in the advice network and decreased in the discussion network. |
| Gesell et al. (2016) [57] | **Design:** Uncontrolled pre-post test **Sample:** Parents of children (3–6 years old) at risk for obesity; n = 305 | Community recreation center in Nashville, TN | Advice and discussion ties with group members outside of training sessions | **Intervention:** In-person group meetings **Intervention summary:** GROW intervention; explained above | **Measures:** network size (number of nominations) in advice and discussion networks **Analysis:** comparison of the average number of nominations at week 3 and 6 | Significant increase (from week 3 to 6) in cohesion and advice nominations; Non-significant increase in discussion nominations; New network nominations were associated with perceptions of group cohesion. |
| Tesdahl et al. (2015) [56] | **Design:** Uncontrolled pre-post test **Sample:** Expectant Latina and African American mothers; n = 59 | Community recreation center in Nashville, TN | People within the program with whom the respondent spoke about pregnancy-related health issues | **Intervention level:** In-person, small groups **Intervention summary:** The intervention included small-group activities aimed at achieving a common goal (e.g., planning an event for family and friends) with rotating leadership roles to increase group cohesion. The intervention also included social skills practice for building and strengthening positive support among family and friends, including identifying existing supports for prenatal health as well as gaps in support networks, identifying the benefits and attributes of supportive relationships, and learning how to build new and tend to supportive relationships. | **Measures:** Participants responded to name generator surveys at weeks 6 and 12 (later mixed together to create a cross-sectional network) to identify other study participants with whom they have spoken about well-being. **Analysis:** The merged cross-sectional network (of weeks 6 and 12) was analyzed using Exponential Random Graph Modeling with tie existence between pairs of participants as the dependent variable, and the total number of sessions the pair attended as the independent variable indicating program effect. | Participants created 3.5 ties on average; Only 4% of ties at week 6 remained the same at week 12; Similarity in physical activity level and due dates increased the likelihood of tie formation; Attendance significantly impacted the formation of network ties among pairs of participants. |

In their qualitative analysis, Osilla et al. (2016) [52] showed that visualization through the development of personal network maps was useful in helping participants build insight on the composition of their networks.

### Group-level network interventions

Interventions in this category were administered in small groups and, in addition to network-building at individual levels, also aimed to facilitate communication, support, and role-modeling among peers within the groups (Table 1).

**Study designs.** Four studies were case reports and uncontrolled pre-post designs. Kasari et al. (2016) [50] conducted an RCT to compare an individual- and a group-level intervention.

**Intervention components.** Group-level interventions involved delivering social skills training [50, 55, 56], or providing opportunities for co-participation in activities of common interest [56]. Two studies involved peer support and training through pairing isolates with highly connected actors [57, 58], while another study encouraged peer support and training through co-participation of individuals and their support workers [55].

Three studies explicitly incorporated the analysis of network structure into their interventions. In van Asselt-Goverts et al. (2018) [55], development of and reflection on personal network maps was a component of group training. Facilitators helped participants develop personal support maps using concentric circles representing tie strength, various social roles (e.g. family, friends, neighbors), and frequency, type, and preference of relations to each alter. However, no alter-to-alter relation data were collected. In two studies [57, 58], the focus and content of group activities were guided by *network* diagnostics that involved the structural analysis of social networks (such as identification of isolates and components/subgroups, and calculation of degree, density, reciprocity, transitivity, centralization, and average of inverse distance). The network diagnostics informed strategies to modify structural characteristics.

**Outcomes.** van Asselt-Goverts et al. (2018) [55] reported the size, frequency, and functional characteristics of social relations to important social roles (family, friends, colleagues, neighbors, others). They also reported various qualitative themes explaining improvement in social connectivity of participants, including awareness, competence, autonomy and participation. Gesell et al. (2013) [58] reported the change in structural measures (e.g. density, reciprocity, clustering, isolates, etc.) over time. They reported a significant increase in the advice network density, an increase in network centralization and in network cohesion, over time. Gesell et al. (2016) [57] was developed based on the experience of pilot assessment in Gesell et al. (2013) [58], and reported significant improvement in the number of nominations in the advice network and subjective measure of cohesion. Kasari et al. (2016) [50] reported social network salience scores (see above). Tesdahl (2015) [56] developed an exponential random graph model to assess how personal, interpersonal, and structural features of the network were associated with the existence of ties between pairs of actors. The model showed that the total number of sessions that pairs of participants attended, as well as similarity in physical activity and pregnancy due date would significantly increase the chance that they are connected in the conversation network. However, they did not directly assess the effect of the intervention, since there were no parallel control groups or longitudinal assessments of network formation.

## Discussion

Our review identified multiple interventions that may mitigate social isolation and loneliness among vulnerable individuals and groups. We describe these programmatic elements, their implications for network interventions generally, and potential applications to address social isolation and loneliness that has been exacerbated by COVID 19 pandemic.

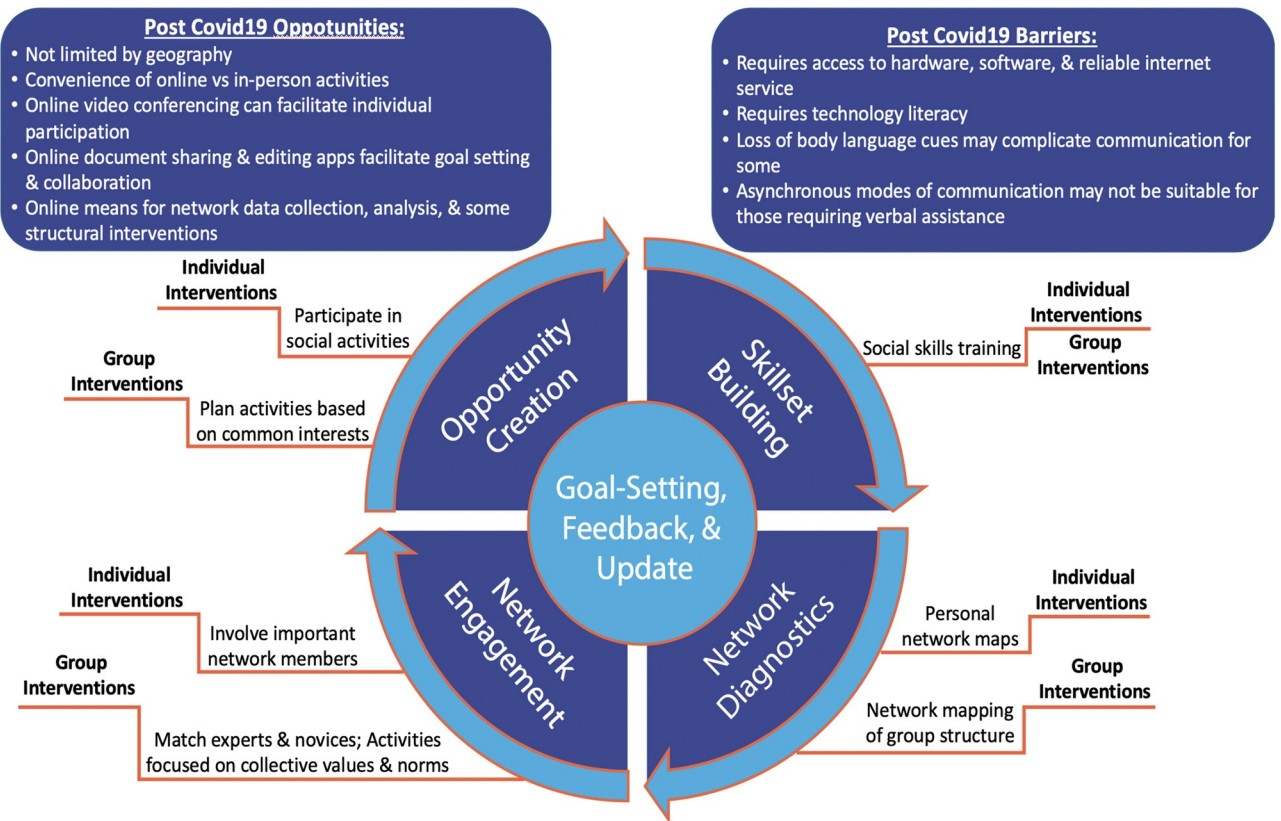

**Fig 2. Components of individual- and group-level network-building interventions; opportunities and barriers for post-COVID19 adaptation.**

## Networks as intervention components

In this review, we identified key elements from both individual-level and group-level network-building interventions that address social isolation and loneliness in a framework shown in Fig 2. These elements, described below, consist of the following: (a) creating opportunities to build networks; (b) building social skills; (c) assessing network diagnostics; and (d) promoting network engagement.

**Creating opportunities for networking.** Both groups of studies (e.g., individual-level and group-level) included interventions that provided opportunities for socialization and networking via participation in events and activities of common interest [50, 51, 53]. When opportunities exist, either in-person or online, this strategy facilitates natural network-building and expansion of the boundaries of social networks to a larger population of individuals with common interests. However, it seems that the mere provision of opportunities was ineffective in forming sustained social ties, and other active strategies are needed.

**Building social skills.** Building social skills through lectures, role modeling, and games were used in some individual- and group-level studies [50, 52, 53, 55–57]. This is particularly useful for individuals who need assistance in building and maintaining social relations. In some studies, the skillset training have been enhanced through peer learning and reflection [55, 56].

**Network diagnostics.** Few studies formally used network structure in the intervention. Studies varied by using network maps as reflective tools and conversation facilitators, [51, 52, 55] to the deliberate use of network analysis as a diagnostic tool [57, 58]. Few social network

interventions in our study made any deliberate attempt to alter network structure. The GROW intervention, with its emphasis on connecting isolates to other group members and connecting clusters together is an exception [57, 58].

**Promoting network engagement.**   Engagement of influential actors in the process of network-building was used in some studies. In individual-level interventions, this was accomplished by inclusion of significant network members (e.g. caregivers, immediate family) in the sessions [50, 53, 57, 58]; and in group-level interventions by matching novices with experts or isolates with central actors [57, 58]. Such strategies formally incorporate social influence and opinion-leadership in the network-building strategy.

**Goal setting, feedback, and update.**   Most of the included studies recognized the gradual process of network-building and the possibility of trial-and-error. Brief and short-term interactions are less likely to lead to sustainable network building. Consequently, incorporating this iterative and reflective process into interventions is an important consideration in designing network-building strategies that applies to all key elements introduced in Fig 2. Setting and updating goals based on feedback allows for individuals and groups to be actively engaged in network-building, leading to more promising network outcomes.

Since structural, social, interpersonal, and intrapersonal factors contribute to social isolation and loneliness [59], different components of network-building interventions (e.g. mapping, self-reflection, and creating networking opportunities) might be helpful within different contexts. Mapping, reflection, and maintenance of existing networks could be more helpful when existing ties have untapped potential. However, when existing networks are limited, overburdened, or lack the capacity to provide needed support, individuals might need to expand their networks by building ties with new contacts.

## Networks as outcomes

Very few studies of interventions to address loneliness/isolation measure the structure of social networks. Even though included studies collected data on the structure of networks, their reported outcomes were mostly limited to basic measures, such as network size, or relative centrality of actors in the network. We did not find any study that assessed if and how interventions would affect the density, clustering, and reciprocity of relations, and whether their effect would focus on certain types of relations, or certain social circles (e.g. family, or intimate relations). Much of the literature assumes that the goal of intervention is to form links with individuals, who may then offer increased social support. This effectively (and sometimes explicitly) draws on a paradigm in which an individual's direct connections to friends and family bring various kinds of social support, most often emotional or instrumental [60]. This is somewhat disappointing, because social network structure matters beyond the simple number of connections. There is evidence that network reciprocity encourages information diffusion [61]; and individuals find clustered networks, those with multiple closed triads, to be more supportive [62]. Clustered social networks, which include closed triads in which three people all know each other, are more supportive and more powerfully influence individuals than networks with more open triads [62, 63]. The quality of social network connections is also of concern [64]. For example, it seems that network structure and quality could be easily addressed in a program such as Genie [51], and based on the authors' description of the program, it appears to be quite feasible to implement through online audio/video platforms such as Skype or Zoom.

Additionally, studies have not addressed the time over which network changes following intervention can be expected to endure. Interventions that strengthen relationships within

existing subgroups, and provide support for their sustainment might be especially promising for reducing isolation.

## Implications for post-COVID19 social relations

COVID19 pandemics and its lock-down and "social distancing" consequences worsened, and yet raised attention to the widespread problem of loneliness and its impacts on mental and physical health, particularly in older adults and vulnerable populations [65–69]. To fight against the spread of COVID-19 in the communities, countries worldwide took measures, such as social distancing, stay-at-home, shelter-in-place orders, and restricting visits to residential facilities (e.g., nursing homes) [70]. The "COVID-19 connectivity paradox" posited that factors improving social connectedness seem to increase the risk of COVID-19 exposure among populations vulnerable to the negative impact of the pandemic as well social isolation [71]. Even with the new hopes invoked by the widespread administration of vaccines, the likelihood of new mutations and upcoming seasonal waves will limit many natural opportunities for face-to-face social communication and network-building for the time being.

Network-building interventions to mitigate loneliness in individuals most in need should consequently be adapted to the new context, and to balance the need for safety and connectedness [71]. Many of these key elements could be implemented online and the trainings and preparations do not seem to be heavily burdensome for health care systems. In this scoping review, we developed a conceptual framework for various potential components of network-building interventions (Fig 2), and discussed various strategies applied in interventional studies to operationalize them. Even though this review does not synthesize the evidence of effectiveness of intervention components and potential synergistic effects of their combinations, we argue that our framework can provide guidance for intervention developers to decisively choose and mix intervention components to develop potentially effective interventions that work, even if some components are not as feasible or as effective as pre-COVID19 conditions. Our review can provide practical insights into the design and implementation of network-building interventions and potential considerations in adapting them to the needs of different target groups. In Fig 2, we summarized potential opportunities and barriers to the implementation of each key element of network-building interventions.

Since the early stages of the COVID19 pandemic and in subsequent months, many health care systems adopted various telehealth care models [72, 73], so individuals connected to institutional or community services still, at least partially, benefit from their ties with professionals and staff virtually. In other words, social workers, home health aides, and other professionals connected to populations at risk of social isolation and loneliness may be well-positioned to conduct network interventions. Their assistance is particularly valuable to people without natural connections within a household (e.g. widowed older adults). For instance, staff and volunteers in programs such as Meals on Wheels serving homebound older adults, are an obvious place to start. Consumers of such formal services could then suggest other peers who might benefit, leading to a snowball sample. Community-based programs have also been found to enhance the neighborhood social networks of older adults, potentially mitigating the risk for social isolation and loneliness [26]. Another possibility to expand social networks is to leverage the skills of paraprofessionals or even volunteers, rather than clinicians, as is done in self-help groups. While the successful Network Support program uses clinicians to facilitate the integration of recovering alcoholics into networks of sober peers [74], twelve-step programs have accomplished the task for decades with and without professional assistance [75]. There are also programs that teach individuals' trusted natural ties, family and friends, to deliver social

network interventions [25, 55]. The phrase, "Each one teach one," could easily be modified to, "Each one reach one."

Although virtual interactions may not completely replace the benefits of in-person contacts, studies demonstrated that older adults who maintained a similar level of virtual social interaction (e.g., social media, phone) with people outside of the household had lower levels of depressive symptoms than those who increased or reduced their virtual interactions during the COVID-19 pandemic [76]. One plausible explanation is that maintaining similar levels of virtual connections allows individuals to receive support without feeling overwhelmed by the stress and worry about the pandemic, which can also be delivered through social networks [76]. Since structural, social, interpersonal, and intrapersonal factors contribute to social isolation and loneliness [59], different components of network-building interventions (e.g. mapping, self-reflection, and creating networking opportunities) might be helpful within different contexts. Mapping, reflection, and maintenance of existing networks could be more helpful when existing ties have untapped potential.

Scalability is an issue with a number of the social network interventions identified in our review. Not surprisingly, a number of the interventions were developed and tested using clinicians, who may be prohibitively expensive during a large-scale crisis. If scale is a challenge for interventions that focus on helping individuals to map and understand their networks, interventions that create groups for isolated individuals can forge connections without needing a facilitator. However, these interventions might be limited during social distancing. One possibility is to create groups online based on common interests. It might also be possible to construct micro-groups of three or four who would meet face to face either regularly or occasionally while practicing safe social distancing [56].

## Limitations

We recognize that the studies included in this review were drawn from a larger study examining social network interventions across all levels (e.g., individuals, groups, communities, organizations). Thus, there may be additional strategies to support network building (e.g., interventions targeting larger communities and organizations) that were omitted from the present findings. Furthermore, due to the dearth of research that examines social network intervention outcomes, we focused on providing a picture of the typology and common elements of network-building interventions, rather than quality appraisal and synthesis of effectiveness, that could enhance the interpretability of our findings, and only relied on authors' statements on the effectiveness of interventions. The majority of these studies relied on uncontrolled research designs limiting our understanding of the effectiveness of these interventions, underscoring the need for more rigorous trials.

## Conclusions

Network-building interventions to address social isolation/loneliness have different combinations of five key elements: (1) creating opportunities for networking and socialization, (2) building social skills, (3) informing the interventions by network analysis of personal and group networks, (4) engagement of influential network members, and (5) goals setting, feedback, and update. The choice of intervention elements is a decision that should be made in light of the nature of the social relations, characteristics of participants, expertise of the facilitators, and contextual factors (such as access to online communication resources, availability of local services, willingness and accessibility of network members). Little has been done to assess how network-building interventions actually change the structure of social networks, beyond simply the number of contacts. Future studies should focus on assessing the effect of

intervention elements and their combinations, and the effect of interventions on network structural outcomes.

In the midst of COVID19 pandemic, we are in urgent need of innovative approaches for building and maintaining social networks among those at risk for social isolation and loneliness. Online interventions or a combination of online, phone, and in-person interventions may facilitate network building among vulnerable individuals and groups. Alternative forms of delivery (e.g. phone or mail) might be helpful for people with limited access to the internet. The main motivation to connect through group-level interventions could vary from commonalities in the neighborhood, demographics, and common health conditions. Given the limited opportunities for group activities in online environments, specific attention should be paid to feasibility testing and adaptation. Attention should be paid to motivating and maintaining social engagement in the group context. Individual and group-level interventions should be delivered in stepwise, iterative, and reflective styles. More studies are needed to identify what combination of network-building elements works best under what conditions. Until very recently it seemed that there would be time to slowly build the evidence base to address the increasing problem of loneliness. The current COVID19 pandemic will hopefully force researchers to speed up that timetable.

## Supporting information

**S1 Appendix. SCOPUS search strategy.**
(DOCX)

## Acknowledgments

The authors thank Elena Navarro for her assistance in data screening and extraction.

## Author Contributions

**Conceptualization:** Reza Yousefi Nooraie, Keith Warren, Lisa A. Juckett, Alicia C. Bunger.

**Formal analysis:** Reza Yousefi Nooraie, Keith Warren, Lisa A. Juckett, Qiuchang A. Cao, Alicia C. Bunger, Michele A. Patak-Pietrafesa.

**Investigation:** Reza Yousefi Nooraie, Keith Warren, Lisa A. Juckett, Qiuchang A. Cao, Alicia C. Bunger, Michele A. Patak-Pietrafesa.

**Methodology:** Reza Yousefi Nooraie, Keith Warren, Lisa A. Juckett, Qiuchang A. Cao, Alicia C. Bunger, Michele A. Patak-Pietrafesa.

**Project administration:** Reza Yousefi Nooraie, Alicia C. Bunger.

**Writing – original draft:** Reza Yousefi Nooraie, Keith Warren, Lisa A. Juckett, Qiuchang A. Cao, Alicia C. Bunger, Michele A. Patak-Pietrafesa.

**Writing – review & editing:** Reza Yousefi Nooraie, Keith Warren, Lisa A. Juckett, Qiuchang A. Cao, Alicia C. Bunger, Michele A. Patak-Pietrafesa.

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
