## [Decision Letter · Decision Letter 0]

5 Mar 2021

PONE-D-20-26180

Individual- and Group-Level Network-Building Interventions to Address Social Isolation and Loneliness; A Scoping Review with implications for COVID19

PLOS ONE

Dear Dr. Yousefi Nooraie,

Thank you for submitting your manuscript to PLOS ONE. After careful consideration, we feel that it has merit but does not fully meet PLOS ONE’s publication criteria as it currently stands. Therefore, we invite you to submit a revised version of the manuscript that addresses the points raised during the review process.

The manuscript has been evaluated by two reviewers, and their comments are available below.

The reviewers are positive about the work but have raised a number of concerns that need attention. They request additional information and reporting on aspects of the study including but not limited to complete summaries of outcomes, effectiveness, and impact, revisions to the statistical analyses and they have requested clarification on the rationale and objectives for the review.

Could you please revise the manuscript to carefully address the concerns raised below?

We look forward to receiving your revised manuscript.

Kind regards,

Vanessa Carels

Staff Editor

PLOS ONE

Journal Requirements:

Reviewers' comments:

Reviewer's Responses to Questions

**Comments to the Author**

1. Is the manuscript technically sound, and do the data support the conclusions?

Reviewer #1: Yes

Reviewer #2: Partly

2. Has the statistical analysis been performed appropriately and rigorously? 

Reviewer #1: N/A

Reviewer #2: N/A

3. Have the authors made all data underlying the findings in their manuscript fully available?

Reviewer #1: Yes

Reviewer #2: Yes

4. Is the manuscript presented in an intelligible fashion and written in standard English?

Reviewer #1: Yes

Reviewer #2: Yes

5. Review Comments to the Author

Reviewer #1: Dear authors

The scoping review is completed in a rigorous and comprehensive manner, and in the most part, conforms to PRISMA guidelines for reporting of scoping reviews. Below are some elements that should be clarified for the readership:

Some minor typos / problems with sentence structure

Page 3

1. (introduction) – do you mean physical and mental health (rather than ‘health and mental health), or particularly mental health?

2. “ and usually provide a single scale”. Do you mean single score? Please clarify.

3. “…in respondent’s social network [25]”. Do you mean in a respondent’s social network OR in respondents’ social networks??

Page 5

4. “We followed PRISMA-ScR guidelines for scoping reviews in the conduct of the literature review, data extraction/charting, and synthesis [48]” – is this clearer?

Page 7

5. “Three studies explicitly incorporated the practice of? analyzing network structure into their interventions. Or ‘…. The analysis of network structure….?

Page 9

6. P ? states “Network structural and quality considerations could easily be implemented in a program such as Genie [54], and are quite amenable to online audio/video platforms such as Skype or Zoom”. Is it their assessment that is amenable to an online platform? Please clarify this sentence.

Other minor clarifications necessary

7. How many reviewers / pairs of reviewers were there? Was there screening for duplicates? Were there any disagreements and how were there resolved – for each stage of the full screening process /study selection?

8. Was a quality appraisal conducted? Was this relevant the eh objectives of the review? There is a mention of quality of studies in the discussion.

9. Summarise what categories of data was extracted from studies (in methods section)

Mores substantial changes necessary:

10. The rationale and particularly the objectives of the review aren’t very clear. This does map onto the requirements of the prisma guidelines. The following statement (p5) indicates which articles are the focus of the scoping review, but what is the point of the review / review questions? Is it to map this literature? To identify specific intervention components?

“We conducted a systematic scoping review of studies that assessed how social network interventions that address loneliness and isolation affect the structure of social networks. We evaluated how these interventions can be adapted to promote connectedness in the wake of COVID19.” (p5)

11. In summarising the results, it is not clear whether the interventions described are in-person or not. Band et al (2019) is described as an online platform, delivered in person. Are all others also in-person? Given the links to the covid context (delivery of virtual / online services), these characteristics are important to include.

12. The outcomes / effectiveness / impact has not been summarised for all the studies mentioned in the narrative. Were they effective / have impact on social networks (or elements of)? For example, in the following narrative, only the outcomes of the Tesdahl (2015) study are described. It may be helpful to summarise briefly the effectiveness (as reported by the authors of the primary papers) of other studies. This is also relevant given the comment made by the review authors that “In the face of COVID 19, it is critical to consider the factors that facilitate or impede the potential use of social network interventions to alleviate social isolation and loneliness”. Why have the results of some of the studies been summarised, but not others, and are there implications around quality of the studies?

“van Asselt-Goverts et al. (2018) [58] reported the size, frequency, and functional characteristics of social relations to important social roles (family, friends, colleagues, neighbors, others). Only Gesell et al. (2013)[60] studied change in network structure and descriptively reported the change in structural measures (e.g. density, reciprocity, clustering, isolates, etc.) over time. Kasari et al. (2016)[53] reported social network salience scores (see above). Tesdahl 8 (2015)[59] developed an exponential random graph model to assess how personal, interpersonal, and structural features of the network was associated with existence of ties between pairs of actor. The model showed that the total number of sessions that pairs of participants attended, as well as similarity in physical activity and pregnancy due date would significantly increase the chance that they are connected in the conversation network. However, they did not directly assess the effect of the intervention, since there were no parallel control group or longitudinal assessment of network formation.”

13. The beginning of the discussion states that “Our review identified multiple interventions that may mitigate social isolation and loneliness among vulnerable individuals and groups”. Related to the point 11 above, it is not clear from the summary of results which interventions were effective. Also, were the interventions included in the review all targeted at vulnerable groups? What types of ‘vulnerable’ groups? Was there any eligibility criteria about the interventions being provided to vulnerable groups?

14. It is important that the review authors provide more context around the covid19 situation (for e.g. isolating, shielding, particularly of vulnerable groups, closure of social / community groups and facilities) so that the reader can understand why specific intervention components are relevant to the covid 19 situation.

15. Related to 15 (above) the context of social distancing / lockdown isn’t fully discussed. One consideration (which the authors have mentioned) is how these intervention can be ‘delivered’ by HCPs, with/without facilitators, or even virtually. But another important consideration is HOW the interventions will result in positive outcomes when (during covid 19) there are more limited options for face-to-face contact (particularly for vulnerable individuals). Are the authors suggesting that the users will develop/improve their networks online /on the phone etc? What are the implications of this for perceived support and loneliness, i.e. does this replicate / approximate in-person networks?

16. Finally, the authors state “In Figure 2, we present potential opportunities and barriers to implementation of each key element of network-building interventions”. It isn’t obvious to me how the figure presents opportunities and barriers to implementation of each key element… Can this be explained further?

Reviewer #2: This is an interesting paper but it could be improved by fleshing out what are the outcomes of the interventions reviewed in relation to social isolation and loneliness. The paper could engage in a more narrative style that highlights the implications of those findings. Some information about the characteristics of the samples in the interventions would be useful as well, as research has showed that people's social networks differ along the life course (i.e. regarding quantity and quality).

Additional comments:

It is not describe how the authors went from 3390 references to assess 233.

p.13 -- Student??. Did the author mean studies rather than students?

There is not much discussion about the effectiveness of the interventions (and for whom were effective).

6. PLOS authors have the option to publish the peer review history of their article (what does this mean?). If published, this will include your full peer review and any attached files.

Reviewer #1: No

Reviewer #2: No

---

## [Author Response · Author response to Decision Letter 0]

11 May 2021

Dear esteemed editors and reviewers of PLOS ONE,

Thank you for thorough assessment of our manuscript, titled “Individual- and Group-Level Network-Building Interventions to Address Social Isolation and Loneliness; A Scoping Review with implications for COVID19”. We tired our best to address reviewers’ comments. Please find attached, the point-by-point response to the reviews. We highlighted the sections that are revised to address suggested revisions, in yellow in the track-changed version of the manuscript.

Looking forward to your positive consideration.

Reza Yousefi Nooraie 

5. Review Comments to the Author

Reviewer #1: Dear authors

The scoping review is completed in a rigorous and comprehensive manner, and in the most part, conforms to PRISMA guidelines for reporting of scoping reviews. Below are some elements that should be clarified for the readership:

Some minor typos / problems with sentence structure

Page 3

1. (introduction) – do you mean physical and mental health (rather than ‘health and mental health), or particularly mental health?

Corrected.

2. “ and usually provide a single scale”. Do you mean single score? Please clarify.

Corrected.

3. “…in respondent’s social network [25]”. Do you mean in a respondent’s social network OR in respondents’ social networks??

Corrected.

Page 5

4. “We followed PRISMA-ScR guidelines for scoping reviews in the conduct of the literature review, data extraction/charting, and synthesis [48]” – is this clearer?

Thank you. Corrected.

Page 7

5. “Three studies explicitly incorporated the practice of? analyzing network structure into their interventions. Or ‘…. The analysis of network structure….?

Corrected.

Page 9

6. P ? states “Network structural and quality considerations could easily be implemented in a program such as Genie [54], and are quite amenable to online audio/video platforms such as Skype or Zoom”. Is it their assessment that is amenable to an online platform? Please clarify this sentence.

This is our interpretation of the intervention, based on the information provided in the included papers. We revised the sentence for clarity.

Other minor clarifications necessary

7. How many reviewers / pairs of reviewers were there? Was there screening for duplicates? Were there any disagreements and how were there resolved – for each stage of the full screening process /study selection?

We calibrated our screening and data extractions on a sample of papers that were independently reviewed by pairs of reviewers. However, information from included studies were extracted by a single reviewer. The group regularly met to discuss and resolve uncertainties and potential disagreements, and continuously updated the data-charting form. We added clarification to this section.

8. Was a quality appraisal conducted? Was this relevant to the objectives of the review? There is a mention of quality of studies in the discussion.

Consistent with the common practice of scoping reviews, we focused on providing a picture of common elements of network-building interventions, rather than quality appraisal and synthesis of effectiveness. 

This section was provided in the Limitations section.

9. Summarise what categories of data was extracted from studies (in methods section)

We added the following to the Methods section:

For each included study, we extracted information pertaining to the study settings, characteristics of network actors, the nature of network relations, characteristics of the intervention(s), theoretical underpinnings of the intervention(s) according to the authors, measures of network change, and the effectiveness of the intervention according to the authors. 

Mores substantial changes necessary:

10. The rationale and particularly the objectives of the review aren’t very clear. This does map onto the requirements of the prisma guidelines. The following statement (p5) indicates which articles are the focus of the scoping review, but what is the point of the review / review questions? Is it to map this literature? To identify specific intervention components?

“We conducted a systematic scoping review of studies that assessed how social network interventions that address loneliness and isolation affect the structure of social networks. We evaluated how these interventions can be adapted to promote connectedness in the wake of COVID19.” (p5)

We revised the aims of the study as follows: “we conducted a systematic scoping review of studies assessing the effect of interventions to address loneliness and social isolation on the structure of social networks. We aimed to map the types and components of these interventions and methods/metrics of assessing changes in social networks.”

11. In summarising the results, it is not clear whether the interventions described are in-person or not. Band et al (2019) is described as an online platform, delivered in person. Are all others also in-person? Given the links to the covid context (delivery of virtual / online services), these characteristics are important to include.

Thank you. We added a section to Table 1, indicating the intervention level.

12. The outcomes / effectiveness / impact has not been summarised for all the studies mentioned in the narrative. Were they effective / have impact on social networks (or elements of)? For example, in the following narrative, only the outcomes of the Tesdahl (2015) study are described. It may be helpful to summarise briefly the effectiveness (as reported by the authors of the primary papers) of other studies. This is also relevant given the comment made by the review authors that “In the face of COVID 19, it is critical to consider the factors that facilitate or impede the potential use of social network interventions to alleviate social isolation and loneliness”. Why have the results of some of the studies been summarised, but not others, and are there implications around quality of the studies?

“van Asselt-Goverts et al. (2018) [58] reported the size, frequency, and functional characteristics of social relations to important social roles (family, friends, colleagues, neighbors, others). Only Gesell et al. (2013)[60] studied change in network structure and descriptively reported the change in structural measures (e.g. density, reciprocity, clustering, isolates, etc.) over time. Kasari et al. (2016)[53] reported social network salience scores (see above). Tesdahl 8 (2015)[59] developed an exponential random graph model to assess how personal, interpersonal, and structural features of the network was associated with existence of ties between pairs of actor. The model showed that the total number of sessions that pairs of participants attended, as well as similarity in physical activity and pregnancy due date would significantly increase the chance that they are connected in the conversation network. However, they did not directly assess the effect of the intervention, since there were no parallel control group or longitudinal assessment of network formation.”

The aim of this scoping review was to provide an overall picture of current research in the fields, and not to synthesize the results of the included studies (as stated in the limitations). However, to provide a preliminary depiction of study findings we added a column to Table 1 indicating the findings, and expanded the ‘outcomes’ section of the Results to provide a summary of the evidence of effectiveness of the intervention as presented in each study. 

13. The beginning of the discussion states that “Our review identified multiple interventions that may mitigate social isolation and loneliness among vulnerable individuals and groups”. Related to the point 11 above, it is not clear from the summary of results which interventions were effective. 

We did not change this section, since we provided more details about the study findings in the Results and Table 1.

Also, were the interventions included in the review all targeted at vulnerable groups? What types of ‘vulnerable’ groups? Was there any eligibility criteria about the interventions being provided to vulnerable groups?

Our eligibility criteria were limited to studies using network interventions to address loneliness/social isolation. We did not intentionally limit the studies to vulnerable populations. But given the nature of the study outcome, most included studies were focused on vulnerable individuals. 

14. It is important that the review authors provide more context around the covid19 situation (for e.g. isolating, shielding, particularly of vulnerable groups, closure of social / community groups and facilities) so that the reader can understand why specific intervention components are relevant to the covid 19 situation.

15. Related to 15 (above) the context of social distancing / lockdown isn’t fully discussed. One consideration (which the authors have mentioned) is how these intervention can be ‘delivered’ by HCPs, with/without facilitators, or even virtually. But another important consideration is HOW the interventions will result in positive outcomes when (during covid 19) there are more limited options for face-to-face contact (particularly for vulnerable individuals). Are the authors suggesting that the users will develop/improve their networks online /on the phone etc? What are the implications of this for perceived support and loneliness, i.e. does this replicate / approximate in-person networks?

We expanded the Discussion and provided further rationale and contextual information for post-COVID social contexts. As indicated in the Discussion: Even though this review does not synthesize the evidence of effectiveness of intervention components and potential synergistic effects of their combinations, we argue that our framework can provide a guidance for intervention developers to decisively choose and mix intervention components to develop potentially effective interventions that work, even if some components are not as feasible or as effective as pre-COVID19 conditions. 

16. Finally, the authors state “In Figure 2, we present potential opportunities and barriers to implementation of each key element of network-building interventions”. It isn’t obvious to me how the figure presents opportunities and barriers to implementation of each key element… Can this be explained further?

Thank you for bringing this up. This statement referred to a more detailed version of the Figure that is now included in the manuscript

Reviewer #2: This is an interesting paper but it could be improved by fleshing out what are the outcomes of the interventions reviewed in relation to social isolation and loneliness. 

The paper could engage in a more narrative style that highlights the implications of those findings. 

We expanded the Discussion to provide a clear rationale for and potential implications of the conceptual framework we developed (presented in Figure 2). The Figure also includes implications and barriers related to post-COVID19 social contexts. 

Some information about the characteristics of the samples in the interventions would be useful as well, as research has showed that people's social networks differ along the life course (i.e. regarding quantity and quality).

Table 1 now includes more detailed description of participant characteristics and demographics.

Additional comments:

It is not describe how the authors went from 3390 references to assess 233.

We added more details to the PRISMA diagram (Figure 1). Given the complexity of the field of network research and the lack of widely used terminologies, we excluded the majority of records at the abstract screening phase due to obvious lack of relevance to social networks (e.g. studying genomic, neural, or animal networks), irrelevance to social isolation/loneliness (e.g. studies of collaboration, advice, or friendship networks), or lack of interventional components (e.g. observational studies of the structure of networks, or their natural emergence over time). 

p.13 -- Student??. Did the author mean studies rather than students?

The two mentioned studies were on school children. We changed the word ‘student’ to ‘individual’ to prevent confusion.

There is not much discussion about the effectiveness of the interventions (and for whom were effective).

The aim of this scoping review was to provide an overall picture of current research in the fields, and not to synthesize the results of the included studies (as stated in the limitations). However, to provide a preliminary depiction of study findings we added a column to Table 1 indicating the findings, and expanded the ‘outcomes’ section of the Results to provide a summary of the evidence of effectiveness of the intervention as presented in each study.

---

## [Editor Report · Decision Letter 1]

14 Jun 2021

Individual- and Group-Level Network-Building Interventions to Address Social Isolation and Loneliness; A Scoping Review with implications for COVID19

PONE-D-20-26180R1

Dear Dr. Yousefi Nooraie,

We’re pleased to inform you that your manuscript has been judged scientifically suitable for publication and will be formally accepted for publication once it meets all outstanding technical requirements.

In the spirit of transparency I want to inform you that I was also one of the reviewers of your manuscript.

Kind regards,

Noleen McCorry

Guest Editor

PLOS ONE
---

## [Editor Report · Acceptance letter]

17 Jun 2021

PONE-D-20-26180R1 

Individual- and group-level network-building interventions to address social isolation and loneliness; a scoping review with implications for COVID19 

Dear Dr. Yousefi Nooraie:

I'm pleased to inform you that your manuscript has been deemed suitable for publication in PLOS ONE. Congratulations! Your manuscript is now with our production department. 

Kind regards, 

on behalf of

Dr. Noleen McCorry 

Guest Editor

PLOS ONE